# ONEBNET: BINARIZED NEURAL NETWORKS USING DECOMPOSED 1-D BINARIZED CONVOLUTIONS ON EDGE DEVICE

## ABSTRACT

It was known that the decomposed 1-D convolutions can replace the spatial 2-D convolutions in several convolutional neural networks (CNNs) for computer vision. However, the proper usage of 1-D convolutions was not shown in the field of binarized CNNs. This paper proposes a new structure called *OneBNet* to maximize the effects of 1-D binarized convolutions, thus producing excellent performance on CPU-based edge devices. To double the effects of adjusting the activation distribution and non-linear activation function, specific layers for BCNNs are doubled by applying them to both $n \times 1$ row-wise and $1 \times n$ column-wise 1-D binarized convolutions. The proposed 1-D downsampling can perform information compression gradually through two 1-D convolutions, which can contribute tremendously to the performance improvement in binarized convolutional neural networks (BCNNs) in our analysis. In the decomposed 1-D binarized convolution, although computational costs are reduced, the number of element-wise non-linear activation functions and learnable bias layers can be doubled, which can be a significant burden. Therefore, we expect that the 1-D binarized convolution is not suitable for all layers, and we present the reason and experimental results proving it. Based on the above assumption and experimental results, we can provide more optimized structure in terms of performance and costs. With ResNet as a backbone, we evaluate the proposed model on several conventional image datasets. In experiments, the proposed model based on ResNet18 achieves 93.4% and 93.6% Top-1 accuracy on the FashionMNIST and CIFAR10 datasets. In the case of training from scratch, the proposed OneBNet based on ResNet18 can produce 63.9% Top-1 accuracy, showing better performance over the state-of-the-art (SOTA) binarized CNNs based on ResNet18. When applying the teacher-student training, 68.4% Top-1 accuracy can be obtained, which overwhelms the existing SOTA BCNNs. With 5% additional delay on a single thread of Raspberry Pi 4, the proposed lightweight model achieves 67.3% Top-1 accuracy on the ImageNet dataset, outperforming the baseline by 1.8%.

## 1 INTRODUCTION

CNNs have made significant advances in various fields of machine learning. SOTA CNNs and vision transformers overcome human ability as their structural complexity increases. However, the sophisticated models exponentially increase computational and storage resource usage. Whereas a huge parallelism in GPUs or neural network accelerators can achieve speedup for complex models, edge devices do not have enough parallelism for accommodating the increasing model complexity. Quantization on edge devices can make a quantized model adopt SIMD (single-instruction multiple-data) instructions. However, the number of multiple issued instructions is limited, and multiply-accumulate operations are still needed on the quantized model. BCNNs quantize both weights and activations into 1 bit in binarized convolutions. Therefore,

multiply-accumulate operations are replaced by bitwise XNOR and bit-counting operations. However, SOTA BCNNs still have significant accuracy drops over their FP32 counterparts.

While 2-D convolutions have been naturally used in computer vision, 1-D convolutions have been adopted for filtering 1-D sequential data. An existing work (Wu et al., 2021) applied decomposed 1-D binarized convolutions to image classification. Compared to $n \times n$ 2-D convolutions, decomposed $n \times 1$ and $1 \times n$ 1-D convolutions can have smaller kernel size, mainly reducing the amount of computation in convolutions. However, its performance (59.9% Top-1 accuracy on the ImageNet (Russakovsky et al., 2015) dataset) had a significant gap from current SOTA BCNNs. In Bannink et al. (2021), although it is noted that the overhead of shortcuts and element-wise operations is not serious in edge devices, the cases using the decomposed 1-D binarized convolutions are not analyzed. Whereas the decomposed 1-D binarized convolutions require additional element-wise operations and more complex downsampling, the decomposition can double the effects of the shortcuts, adjustments of activation distribution, and non-linear activation functions. There are the above pros and cons when applying the decomposed 1-D binarized convolutions.

Therefore, we propose a new model called *OneBNet*, optimizing the usage of the decomposed 1-D binarized convolutions in BCNNs. The proposed model can contain $n \times 1$ row-wise and $1 \times n$ column-wise 1-D binarized convolutions for image classification. To overcome the weakness of BCNNs using 1-D binarized convolutions, our contributions are as follows:

1. We propose basic blocks using 1-D binarized convolutions with and without downsampling. In the proposed basic block, the row-wise and column-wise downsamplings are gradually performed, which significantly enhances the classification accuracy. The decomposed 1-D binarized convolution using $n \times 1$ row-wise and $1 \times n$ column-wise 1-D convolutions can slightly reduce the amount of computations without significant performance degradation. Besides, we provide the detailed structure of the basic block that adjusts the activation distribution in row-wise and column-wise manners.

2. In order to demonstrate the effectiveness of the proposed basic blocks, we evaluate several different structures based on baseline ResNet (He et al., 2016) models. In the pyramid structure, we conclude that 1-D binarized convolutions can be more effective in deep convolutional layers, considering both increasing computations in downsampling and reduced computations. For the proposed model, we describe the detailed process of training environments and hyperparameters, including the teacher-student training (Hinton et al., 2015) on the ImageNet dataset (Russakovsky et al., 2015).

3. Along with the performance evaluations, the latencies of the different structures are compared on Raspberry Pi 4 using Larq (Bannink et al., 2021), which proves that the proposed structure does not need significant additional latency, compared with baseline ReActNet (Liu et al., 2020).

The proposed model based on ResNet18 achieves 93.4% and 93.6% Top-1 accuracy on the FashionM-NIST (Xiao et al., 2017) and CIFAR10 (Krizhevsky et al., 2014) datasets. Moreover, when adopting the teacher-student training, the proposed OneBNet based on ResNet18 reaches up to 68.4% Top-1 accuracy on the ImageNet dataset. Compared with FP32 ResNet18, the proposed OneBNet can have only 1.2% Top-1 accuracy drop, having only 13% storage costs with $\times 4.7$ inference speed.

## 2 RELATED WORKS

The concept of BCNNs was proposed in Courbariaux et al. (2016). It applied BCNNs on several small datasets. XNOR-Net (Rastegari et al., 2016) shows the validity of BCNNs on the ImageNet dataset. Its binarized ResNet18 (He et al., 2016) scaled the convolution outputs and achieved 51.4% Top-1 accuracy. However, it was about 18% lower than FP32 ResNet18. ABC-Net (Lin et al., 2017) compensated large quantization errors by linearly combining several binary weights. However, both computational and storage

costs increased with the number of combined blocks. Bi-RealNet (Liu et al., 2018) proposed the single skipped connection for each binarized convolutional layer and a customized backward function for the $sign$ function. However, its Top-1 accuracy was only 56.4%, still showing a significant gap from FP32 ResNet18. Real-to-Bin (Martinez et al., 2019) adopted self-attention blocks and shows 65.4% Top-1 accuracy with binarized ResNet18 on the ImageNet dataset. ReActNet(Liu et al., 2020) proposed adjusting the activation distribution through learnable biases, resulting in achieving 65.5% Top-1 accuracy. In Liu et al. (2020); Martinez et al. (2019), the effectiveness of teacher-student training (Hinton et al., 2015) was empirically proved, having over 65% Top-1 accuracy. IR-Net (Qin et al., 2020) minimized information loss in the forward path by maximizing the information entropy and minimizing quantization errors. In SI-BNN (Wang et al., 2020), a trainable threshold was applied to the BCNN to guide the gradient propagation. BNSC-Net (Wu et al., 2021) applied 1-D convolutions to reduce computational complexity. However, BNSC-Net degraded 10% Top-1 accuracy compared to FP32 ResNet18. RB-Net (Liu et al., 2022) simplified operations by reshaping convolutions and applying balanced activations. SA-BNN (Liu et al., 2021) proposed using independent gradient coefficients for different states when updating the weights. AdaBin (Tu et al., 2022) approximated FP32 value distributions through the adaptive binary set and improved the representation of binarized features. DIR-Net (Qin et al., 2023) proposed a distribution-sensitive information-retention network to hold forward activation and backward gradient information. The existing works (Liu et al., 2021; Tu et al., 2022; Qin et al., 2023) mainly proposed better training methods; new BCNN structures were not considered.

PokeBNN (Zhang et al., 2022) significantly enhanced the accuracy of BCNNs. However, the model adopted various types of quantization formats (1-bit, 4-bit, and 8-bit) and self-attention blocks. QuickNet (Bannink et al., 2021) can improve inference speed by using depthwise convolutions in the first convolutional layer. Because the purpose of this paper is to present an efficient method to use the proposed basic block, only FP32 and binary formats are adopted like many BCNNs. Besides, the first and last layers are from the FP32 layers from the baseline ResNet. The number of channels in the proposed model follows the baseline model.

## 3   ONEBNET USING DECOMPOSED 1-D BINARIZED CONVOLUTIONS

### 3.1   MOTIVATIONS

Most of the BCNNs have been developed based on 2-D convolutions. However, when only the order of blocks was changed in Rastegari et al. (2016), its performance was not acceptable, compared to its FP32 baseline. In Bi-RealNet (Liu et al., 2018), the single skipped connection was proposed to apply skipped connection for each binarized convolutional layer. The authors of Bi-RealNet explained that the single skipped connection increased the resolution of binarized convolutions. The single skipped connection can provide better model optimization by preventing internal covariant shift from large quantization errors in the binarized convolutions. The experiments in Bannink et al. (2021) on edge devices showed that the increasing latency of the shortcuts is very small, unlike the cases using GPUs. Therefore, it is concluded that the single skipped connections can be effective in BCNNs. On the other hand, we think that the adjustment of activation distribution and non-linear activation function are essential in BCNNs. For example, RSign and RPReLU of ReActNet (Liu et al., 2020) introduced learnable biases for both inputs and outputs of binarized convolutions. The effects of the learnable biases showed remarkable performance enhancements in BCNNs. We expect that the performance of BCNNs can be enhanced if the above valuable effects over BCNNs can be reinforced.

To double the effects of the adjustment of activation distribution, we apply the adjustment of activation distribution to the decomposed 1-D binarized convolutions. Whereas the conventional 2-D convolutions use $3 \times 3$ kernels, the decomposed convolutions use $3 \times 1$ and $1 \times 3$ kernels. The adjustment of activation distribution is applied to each 1-D convolution. Although the decomposition had the same receptive field with 2-D convolution (Szegedy et al., 2016), it requires additional element-wise operations for the adjustments and activation functions. This paper explains the structure of 1-D convolutional layer. Then, it analyzes the

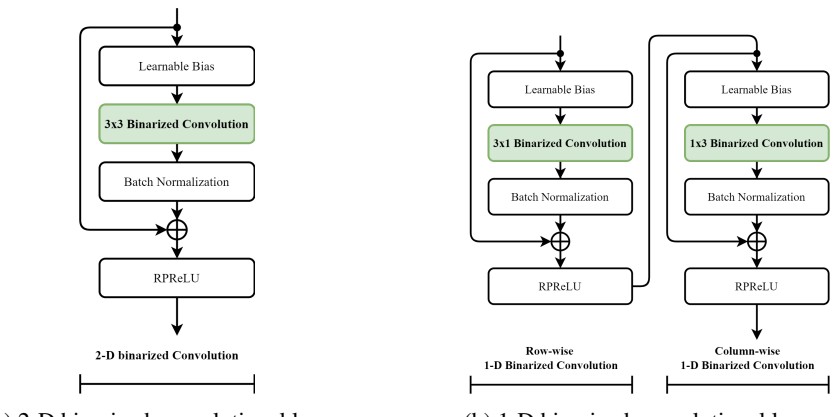

(a) 2-D binarized convolutional layer  (b) 1-D binarized convolutional layers

Figure 1: Spatial convolutions using 2-D and 1-D binarized convolution layers. The spatial convolution using 1-D convolutional layers performed $3 \times 1$ and $1 \times 3$ binarized convolutions.

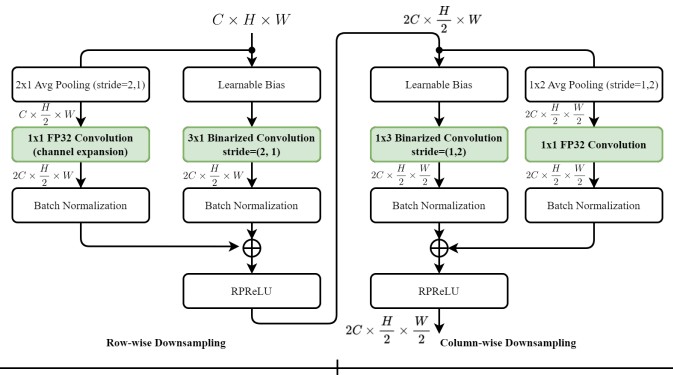

Figure 2: Downsampling using 1-D binarized convolutions. Activations are gradually downsampled in row-wise and column-wise manners using $2 \times 1$ and $1 \times 2$ average poolings (denoted as *Avg Pooling*).

model structures in terms of latency and accuracy and shows the idea to deploy 1-D binarized convolutions, considering the above explained pros and cons.

### 3.2 PROPOSED BASIC BLOCK

The proposed model can have 1-D binarized convolutional layers, which is used as a basic block in the proposed OneBNet. Figure 1 illustrates 2-D and 1-D binarized convolutional layers without downsampling. A symbol $\oplus$ denotes the element-wise addition with a shortcut. Whereas 2-D binarized convolutional layer has $3 \times 3$ binarized convolutional layer in Fig. 1 (a), $3 \times 1$ and $1 \times 3$ binarized convolutions are adopted in Fig. 1 (b), doubling the effects of shortcuts. We think that the two 1-D binarized convolutional layers can correspond to one 2-D binarized convolutional layer. As shown in Liu et al. (2020), activations are adjusted in a learnable bias and RPReLU. Because the learnable bias and RPReLU are used in each row-wise and column-wise convolutions, the numbers of the adjustments of activation distribution and non-linear activation functions are doubled.

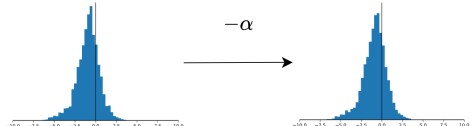 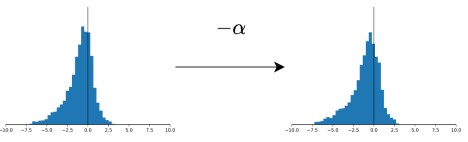

(a) Adjustment of input activations for row-wise binarized convolution

(b) Adjustment of input activation for column-wise binarized convolution

Figure 3: Adjustments of activation distribution with learnable bias in row-wise and column-wise manners.

Figure 2 illustrates the convolutions during downsampling, where terms $C$, $H$, and $W$ denote the number of channels, height, and width in activations. The row-wise 1-D binarized convolution is performed, and then the convolution outputs are summed with the outputs of the expanded channels from the first $1 \times 1$ FP32 convolution, producing $2C \times \frac{H}{2} \times W$ activations. The input activations are adopted in $1 \times 3$ binarized convolution of the column-wise downsampling. Besides, $1 \times 1$ FP32 convolution is also used in the shortcut. The downsampling can gradually perform the information compression and adjustment of activation distribution, producing $2C \times \frac{H}{2} \times \frac{W}{2}$ activations. The $2 \times 1$ and $1 \times 2$ average poolings are performed with $stride = (2, 1)$ and $stride = (1, 2)$. Because two $1 \times 1$ FP32 convolutions are required on the baseline ResNet, the downsampling using 1-D binarized convolutions requires additional computational costs. However, the downsampling is not performed in all basic blocks. For example, there are only three downsampling in binarized convolutions in the baseline ResNet18. Besides, whereas the number of channels doubles, both height and width of activations halved, reducing the number of activations by half. The above characteristics related to computational costs and model performance are used to decide the deployment of 1-D binarized convolutions.

### 3.3 ADJUSTMENT OF ACTIVATION DISTRIBUTION

In (Liu et al., 2020; 2022; Tu et al., 2022), the activation distribution in BCNNs significantly affects model performance. The decomposed 1-D binarized convolutions double the number of the learnable bias and RPReLU layers, reinforcing the effects of adjusting the activation distribution. The equations of the learnable bias and RPReLU for the $i$-th channel are equations (1) and (2) as follows:

$$Learnable\ Bias(x_i) = x_i - \alpha_i. \tag{1}$$

$$RPReLU(x_i) = \begin{cases} x_i - \gamma_i + \zeta_i & \text{if } x_i > \gamma_i \\ \beta_i(x_i - \gamma_i) + \zeta_i & \text{if } x_i \leq \gamma_i \end{cases}. \tag{2}$$

Terms $\alpha_i$, $\beta_i$, $\gamma_i$, and $\zeta_i$ are the learnable parameters for the $i$-th channel. For row-wise and column-wise convolutions, their learnable parameters are adopted to adjust the activation distribution, respectively. Equation (2) formulates the operation of RPReLU, which adjusts the input distribution of PReLU with $\gamma_i$. When $x_i - \gamma_i$ is negative, learnable parameter $\beta_i$ scales it. Figure 3 shows the adjustment of activation distribution with $\gamma_i$ in a basic block using 1-D binarized convolutions. In Fig. 3 (a) and (b), the input distribution of the learnable bias layer is illustrated in the left; the right distributions denote the output distributions of the learnable bias layer. Although the distribution shift is not large, we observe that each distribution is slightly shifted, showing that the learnable bias and RPReLU layers of each row-wise and column-wise 1-D binarized convolution can double the adjustment of activation distribution.

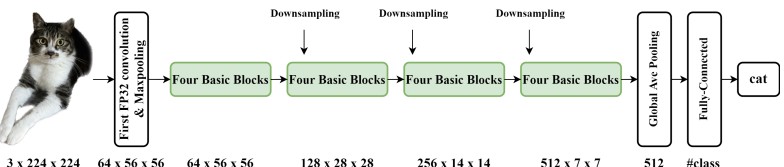

Figure 4: Architecture based on baseline ResNet18. There are sixteen basic blocks, where the layer in Fig. 2 can be deployed during downsampling. The below numbers denote $\#channels \times height \times width$ of activations. The first convolutional and final fully connected layers use FP32 operations like the baseline and conventional BCNNs. More detail description of the model structure is listed in Appendix A.1.

### 3.4 BINARIZED CONVOLUTION

In binarized convolution, both binarized weights are stored, and activations are binarized using the $sign$ function. If a feature or weight is zero, it is binarized as $-1$. The binarized $-1$ and $+1$ values are represented as $0$ and $1$ bits, respectively. To perform multiply-accumulate function in parallel, binarized features and weights are bitwise-XNORed. Then, the accumulation is calculated by counting $+1$ output bits in forward paths. The output of a binarized convolution are multiplied by a FP32 scaling factor shown in Rastegari et al. (2016). The learnable parameters in the next batch normalization layer are used to scale the convolution outputs and add a bias to them. As shown in Bannink et al. (2021), the binarized convolution and batch normalization can be fused by reparameterizing the scaling parameters and biases. During model training, weights in binarized convolution layers are binarized in the forward pass. In backward propagation, FP32 weights are updated based on the training method in Rastegari et al. (2016). Whereas the sign function for binarizing activations is easily implemented, its derivative contains the delta function (Hassani, 2009), which needs its approximate implementation. The derivative of the sign function was approximated into a straight-through-estimator (Yin et al., 2019; Bengio et al., 2013).

### 3.5 MODEL ARCHITECTURE USING 1-D BINARIZED CONVOLUTIONS

Figure 4 illustrates the proposed OneBNet based on the baseline ResNet18. Because the proposed OneBNet follows the structure from ResNet, it is easy to make the pyramid structure using the proposed basic blocks. If all basic blocks adopted 1-D binarized convolutions, there are 32 1-D binarized convolution layers. To obtain the classification output, a 2-D global average pooling layer is deployed. Then, the fully connected layer produces the classification output.

In order to enhance performance without significant speed degradation, computational costs of a basic block are ideally calculated to determine whether 1-D binarized convolutional layers are deployed or not. Figure 5 illustrates the comparison of computations with the cases using 1-D and 2-D binarized convolutions. Terms FLOPs and BOPs denote that the numbers of FP32 and binarized operations, respectively. Each binarized multiplication or addition is calculated as one BOP. Terms OPs are calculated by $OPs = FLOPs + \frac{BOPs}{64}$ (Liu et al., 2020). Figure 5 (a) shows that computations can be reduced with 1-D binarized convolutional layers when the number of channels is 256 or 512. On the other hand, additional OPs in downsampling can affect the inference speed. Therefore, if the proposed 1-D convolutional layers in downsampling cannot provide outstanding enhancements, the usage of 1-D binarized convolutions is meaningless. Our experiments showed that when all downsampling layers use 1-D binarized convolutions, the model can show only 1.2% accuracy drop on the ImageNet dataset, compared with FP32 ResNet18. Besides, when the final downsampling layer uses only 1-D binarized convolutions, it can produce 1.8% accuracy enhancement over the baseline ReActNet18, having small additional latency.

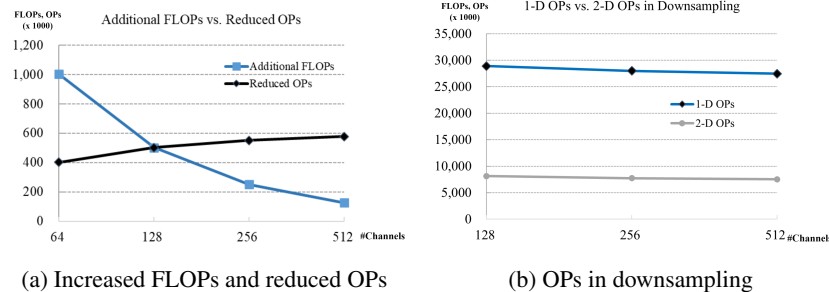

(a) Increased FLOPs and reduced OPs                    (b) OPs in downsampling

Figure 5: Comparison of computations with the cases using 1-D and 2-D binarized convolutions. (a) illustrates the increased FLOPs and reduced OPs of an 1-D binarized convolution compared with a 2-D binarized convolution. It shows that the computations can be reduced by increasing the number of channels. On the other hand, the proposed 1-D binarized convolutions in downsampling require large additional costs.

Table 1: Summary of Top-1 accuracies and latencies by varying structures on ImageNet dataset.

| 1-D Conv | 1-D DS | Top-1 (%) | Latency (ms) | 1-D Conv | 1-D DS | Top-1 (%) | Latency (ms) |
|---|---|---|---|---|---|---|---|
| - | - | 65.5 | 83.2 | 64,128,256,512 | - | 65.4 | 85.5 |
| 256,512 | - | 65.7 | 80.8 | 64,128,256,512 | 128,256,512 | 67.7 | 105.4 |
| 64,128 | 128,256,512 | 68.1 | 106.9 | 256,512 | 128,256,512 | 68.1 | 99.5 |
| - | 128,256,512 | $68.4^{(1)}$ | 101.7 | 64,128 | 128 | 65.8 | 96.3 |
| 256,512 | 128 | 66.2 | 87.7 | 64,128 | 256,512 | 67.6 | 100 |
| 256,512 | 256,512 | $67.7^{(2)}$ | 92.3 | 512 | 512 | $67.3^{(3)}$ | 87.4 |

$^{(1),(2),(3)}$: superscripts$^{(1),(2),(3)}$ denote the models called TypeI, TypeII, and TypeIII depending on the usage of 1-D binarized convolutions. The latency was evaluated on a single thread of Raspberry Pi 4.

## 4 EXPERIMENTAL RESULTS AND ANALYSIS

Experiments were performed on the FashionMNIST, CIFAR10, and ImageNet datasets, where their environments and evaluations on FashionMNIST and CIFAR10 datasets are explained in Appendix A.2 and A.3 in detail. Firstly, we experimented with the proposed OneBNet from ResNet18 on the ImageNet dataset. We followed the two-stage training method and hyperparameters of ReActNet (Liu et al., 2020). In the training method, a teacher-student training method (Hinton et al., 2015) was adopted using Pytorch official pre-trained FP32 ResNet34 as a teacher. In the first stage having 256 epochs, only input features for 1-D convolutions were binarized; weights for 1-D convolutions were FP32 values. In the second stage, the pre-trained weights from the first stage were used in the initialization. Both input features and weights for 1-D convolutions were binarized during 256 training epochs.

Table 1 summarizes Top-1 accuracies and latencies by varying structures. Term **1-D Conv** means the number of output channels of 1-D binarized convolutional layers. Term **1-D DS** denotes the number of output channels of 1-D binarized convolutional layers in downsampling. If 1-D binarized convolutions are not used, they are not listed in Table 1. Compared with 65.5% Top-1 accuracy of ReActNet, when not using 1-D binarized convolutions in downsampling, models cannot show outstanding improvements. In these cases, when 1-D Conv=256,512, its performance was slightly better than other models using 1-D binarized convolutions without downsampling. On the other hand, when three downsamplings adopted 1-D binarized convolutions,

Table 2: Comparison with existing BCNNs on ImageNet dataset.

| Training | Model | Top-1 (%) | Top-5 (%) | Model | Top-1 (%) | Top-5 (%) |
|---|---|---|---|---|---|---|
| Scratch | FP32 ResNet18 | 69.6 | 89.2 | ABC-Net | 42.7 | 67.6 |
| | XNOR-Net | 51.2 | 73.2 | Bi-RealNet | 56.4 | 79.5 |
| | XNOR-Net++ | 57.1 | 79.9 | SI-BNN | 59.7 | 81.8 |
| | IR-Net | 58.1 | 80.0 | BNSC-Net | 59.9 | 81.8 |
| | SA-BNN | 61.7 | 82.8 | RB-Net | 63.0 | 84.2 |
| | AdaBin | 63.1 | 84.3 | **TypeII** | **63.9** | **84.5** |
| TS | Real-to-Bin | 65.4 | 86.2 | ReActNet18 | 65.5 | - |
| | AdaBin | 66.4 | 86.5 | DIR-Net2 | 66.5 | 87.1 |
| | RB-Net | 66.8 | 87.1 | **TypeI** | **68.4** | **88.0** |
| | **TypeII** | **67.7** | **87.7** | **TypeIII** | **67.3** | **87.9** |

Top-1 accuracies were enhanced by 1.8-2.9%. When 1-D binarized convolutions were not adopted for downsampling in TypeI, its latency was 101.7 ms. When 1-D Conv=64,128 and 1-D Conv=256,512, its latencies were 106.9 ms and 99.5 ms, respectively. The difference showed that 1-D binarized convolutions without downsampling can have small benefits in terms of inference speed, as shown in Fig. 5 (a). On the other hand, when 1-D DS=256, 512 and 1-D DS=512, Top-1 accuracies were significantly enhanced, compared with the model with 1-D DS=128. When 1-D Conv=1-D DS=512, Top-1 accuracies can be 67.3% with the latency of 87.3 ms. Table 1 shows the effectiveness of 1-D binarized convolutions in downsampling, where deep convolutional layers with many channels and small activation map can have significant benefits for enhancing performance.

Table 2 shows the comparison with existing BCNNs on the ImageNet dataset. Except for FP32 ResNet18, only binarized models were compared. For apples-to-apples comparison, we trained the proposed TypeII model from scratch during 256 epochs. During training, binarized CNNs based on ResNet18 were trained with 256 mini-batch sizes. Initial learning rate $Ir_{base}$ was set as $1e-3$ with zero weight decaying. In Table 2, the models and their accuracies above the midline were based on the training from scratch, where the accuracies of ResNet18 (He et al., 2016), ABC-Net (Lin et al., 2017), XNOR-Net (Rastegari et al., 2016), Bi-RealNet (Liu et al., 2018), XNOR-Net++ (Bulat et al., 2019), SI-BNN (Wang et al., 2020), IR-Net (Qin et al., 2020), BNSC-Net (Wu et al., 2021), SA-BNN (Liu et al., 2021), RB-Net (Liu et al., 2022), AdaBin (Tu et al., 2022) are listed. Other models below the midline adopted teacher-student training, including Real-to-Bin (Martinez et al., 2019), ReActNet18 (Liu et al., 2020), DIR-Net2 (Qin et al., 2023), which is denoted as *TS*. The models and data marked with bold text are the proposed TypeI, TypeII, and TypeII models and their training results. On the training from scratch, the proposed TypeII reached 63.9% Top-1 accuracies, outperforming other counterparts trained from scratch. When the two-stage based teacher-student training was applied, the proposed TypeI, TypeII, and TypeIII achieved 68.4%, 67.7%, and 67.3% Top-1 accuracies. As far as we know, it overwhelmed the performance of other SOTA BCNNs based on ResNet18. Notably, the proposed model outperformed SOTA BCNNs using reshaped convolutions (Wu et al., 2021; Liu et al., 2022). Compared with RB-Net (Liu et al., 2022), Top-1 accuracy of TypeI was enhanced by 1.6%. Besides, TypeI only shows 1.2% Top-1 accuracy drop compared with FP32 ResNet18 model.

Table 3 summarizes the comparison in terms of storage costs and inference speed. We note that the models in Table 3 were based on ResNet18. Compared with FP32 ResNet18, the proposed TypeI, TypeII, and TypeIII reduced storage costs and inference latency by 87.9%-89.2% and 78.9%-81.9%, respectively. Compared with ReActNet18, the proposed TypeI increased storage costs and latency by 33% and 22%, but it enhanced Top-1

Table 3: Comparison of storage costs and inference latency on ImageNet dataset

| Model[1] | Storage (MB) | Latency (ms) | Top-1 (%) | Model | Storage (MB) | Latency (ms) | Top-1 (%) |
|---|---|---|---|---|---|---|---|
| FP32 ResNet18 | 44.59 | 481.1 | 69.6 | XNOR-Net | 4.00 | 80.7 | 51.2 |
| Bi-RealNet | 4.00 | 80.2 | 56.4 | Real-to-Bin | 5.13 | 97.1 | 65.4 |
| ReActNet18 | 4.05 | 83.2 | 65.5 | **TypeI** | **5.39** | **101.7** | **68.4** |
| **TypeII** | **5.03** | **92.3** | **67.7** | **TypeIII** | **4.82** | **87.3** | **67.3** |

[1] : Because the models are based on baseline ResNet18, the first FP32 convolutional and fully connected layers are the same. Therefore, the difference between models is produced from the structure of binarized convolutional layers.

Table 4: Comparison with existing BCNNs on CIFAR10 dataset.

| Model(W/A) | Top-1(%) | Model(W/A) | Top-1(%) |
|---|---|---|---|
| ResNet18(32/32) | 95.5 | XNOR-Net++(1/1) | 90.2 |
| Bi-RealNet(1/1) | 89.1 | IR-Net(1/1) | 91.5 |
| RAD(1/1) | 90.5 | ReCU(1/1) | 92.8 |
| RBNN(1/1) | 92.2 | ReActNet(1/1) | 92.3 |
| AdaBin(1/1) | 93.1 | DIR-Net(1/1) | 92.8 |
| **TypeI(1/1)**[1] | **93.6** | **TypeII(1/1)**[2] | **93.3** |

[1], [2] : TypeI and TypeII are described in Table 2.

accuracy by 2.9%. Besides, Type II and Type III offered the option of having a lower storage cost and smaller latency. TypeIII enhanced Top-1 accuracy by 1.8%, having only 5% additional latency.

Top-1 accuracies from ResNet20 on the FashionMNIST dataset can reach up to 93.4% and 93.2%, where the accuracy drops were under 0.5%, compared with FP32 ResNet20 model. Table 4 lists Top-1 accuracies of ResNet18, XNOR-Net, Bi-RealNet (Liu et al., 2018), IR-Net (Qin et al., 2020), RAD (Ding et al., 2019), ReCU (Xu et al., 2021), RBNN (Lin et al., 2020), ReActNet18, AdaBin, and DIR-Net on the CIFAR-10 dataset. The proposed model outperformed all other methods. Based on the above performances, We conclude that TypeI and TypeII can achieve good accuracies on the FashionMNIST and CIFAR10 datasets.

## 5 CONCLUSION

This paper proposes new BCNNs for image classification by decomposing binarized 2-D convolutions. In Wu et al. (2021), naively reshaped 1-D BCNNs cannot outperform SOTA 2-D BCNN models. However, compared with 2-D BCNNs, the proposed OneBNet has significant benefits by having higher classification accuracy. This paper shows that 1-D binarized convolutions can be suitable for image classification. The proposed OneBNet significantly outperforms other SOTA BCNN models on the ImageNet dataset using the decomposed 1-D convolutions in downsampling Above all, it showed only 1.2% Top-1 accuracy drop having $\times 4.7$ inference speed, compared with its FP32 baseline. It suggests the possibility that BCNNs can be applied with high performance and acceptable computational efficiency in any edge applications. Notably, the 1-D binarized convolutions could be suitable for implementing CNNs on power-hungry edge devices. Considering the above outperforming experimental results and structural benefits, we assure the proposed OneBNet is the best BCNN model so far. Additional explanations and visualizations for proving the effectiveness of OneBNet are included in the Appendix.

## 6 REPRODUCIBILITY STATEMENT

We adopted conventional FashionMNIST, CIFAR10, and ImageNet datasets for easy reproduction. The attached code can be run when Pytorch dataset formats are prepared. The detail structure when 1-D binarized convolutions is described in Appendix A.1. For better understanding reproduction, additional description of experimental environments and experiments on small datasets are included in Appendix A.2 and A.3. Besides, the environments for evaluating inference speed are described in Appendix A.4. The operation of a basic block can be easily understood based on Appendix A.5. The visualizations of training and internal activations in Appendix A.6 and A.7 could be helpful for understanding the proposed model.

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

# A APPENDIX

## A.1 ARCHITECTURE OF ONEBNET

Table 5: Architecture of OneBNet on ImageNet and CIFAR datasets

| Layer Name | Output Size | ImageNet (Russakovsky et al., 2015) | | Output Size | CIFAR (Krizhevsky et al., 2014) | |
|---|---|---|---|---|---|---|
| | | ResNet18 | ResNet20 | | ResNet18 | ResNet20 |
| Conv1 | 112×112 | 7×7, 64, stride 2 | 7×7, 16, stride 2 | 32×32 | 3×3, 64, stride 1 | 3×3, 16, stride 1 |
| Conv2_x | 56×56 | 3×3 max pool, stride 2 | | 32×32 | - | |
| | | $\begin{bmatrix} 3{\times}1, 64 \\ 1{\times}3, 64 \\ 3{\times}1, 64 \\ 1{\times}3, 64 \end{bmatrix} \times 2$ | $\begin{bmatrix} 3{\times}1, 16 \\ 1{\times}3, 16 \\ 3{\times}1, 16 \\ 1{\times}3, 16 \end{bmatrix} \times 3$ | | $\begin{bmatrix} 3{\times}1, 64 \\ 1{\times}3, 64 \\ 3{\times}1, 64 \\ 1{\times}3, 64 \end{bmatrix} \times 2$ | $\begin{bmatrix} 3{\times}1, 16 \\ 1{\times}3, 16 \\ 3{\times}1, 16 \\ 1{\times}3, 16 \end{bmatrix} \times 3$ |
| Conv3_x | 28×28 | $\begin{bmatrix} 3{\times}1, 128 \\ 1{\times}3, 128 \\ 3{\times}1, 128 \\ 1{\times}3, 128 \end{bmatrix} \times 2$ | $\begin{bmatrix} 3{\times}1, 32 \\ 1{\times}3, 32 \\ 3{\times}1, 32 \\ 1{\times}3, 32 \end{bmatrix} \times 3$ | 16×16 | $\begin{bmatrix} 3{\times}1, 128 \\ 1{\times}3, 128 \\ 3{\times}1, 128 \\ 1{\times}3, 128 \end{bmatrix} \times 2$ | $\begin{bmatrix} 3{\times}1, 32 \\ 1{\times}3, 32 \\ 3{\times}1, 32 \\ 1{\times}3, 32 \end{bmatrix} \times 3$ |
| Conv4_x | 14×14 | $\begin{bmatrix} 3{\times}1, 256 \\ 1{\times}3, 256 \\ 3{\times}1, 256 \\ 1{\times}3, 256 \end{bmatrix} \times 2$ | $\begin{bmatrix} 3{\times}1, 64 \\ 1{\times}3, 64 \\ 3{\times}1, 64 \\ 1{\times}3, 64 \end{bmatrix} \times 3$ | 8×8 | $\begin{bmatrix} 3{\times}1, 256 \\ 1{\times}3, 256 \\ 3{\times}1, 256 \\ 1{\times}3, 256 \end{bmatrix} \times 2$ | $\begin{bmatrix} 3{\times}1, 64 \\ 1{\times}3, 64 \\ 3{\times}1, 64 \\ 1{\times}3, 64 \end{bmatrix} \times 3$ |
| Conv5_x | 7×7 | $\begin{bmatrix} 3{\times}1, 512 \\ 1{\times}3, 512 \\ 3{\times}1, 512 \\ 1{\times}3, 512 \end{bmatrix} \times 2$ | - | 4×4 | $\begin{bmatrix} 3{\times}1, 512 \\ 1{\times}3, 512 \\ 3{\times}1, 512 \\ 1{\times}3, 512 \end{bmatrix} \times 2$ | - |
| | 1×1 | GAP 2-D, FC 1000-D, Softmax | | 1×1 | GAP 2-D, FC 10 or 100-D, Softmax | |

The overall architecture of the proposed OneBNet follows ResNet (He et al., 2016). In this table, all binarized convolutional layers are assumed to have the 1-D structure. Notably, FP32 Conv1 convolutional, the last 2-D global average pooling (denoted as **GAP 2-D**), and fully connected layers (denoted as **FC**) are the same as those of the original ResNet. In Table 5, FashionMNIST having $28 \times 28$ input image is not considered. When ResNet20 is adopted on the FashionMNIST dataset, the output size can be downsampled into $(28 \times 28)$-$(14 \times 14)$-$(7 \times 7)$ three times. The detail information of the stacked 1-D convolutional layers inside the basic blocks are shown in brackets of Table 5, where the height, width, and the number of output channels are listed.

## A.2 EXPERIMENTAL ENVIRONMENTS

We adopted ResNet-based BCNN topology to evaluate the proposed OneBNet in terms of image classification accuracy. Like other BCNN models, the first convolutional and last fully-connected layers adopted FP32 weights and activations. For apple-to-apple comparison, we adopted ADAM (Kingma & Ba, 2014) optimizer in all cases, having $\beta_1 = 0.9$ and $\beta_2 = 0.999$. When training during $E_{epochs}$ epochs, the initial learning rate $lr_{base}$ was assigned. During training, learning rate $lr$ in the $e_{epochs}$-th epoch was decreased based on *poly* policy, which limits the maximum learning rate of the ADAM optimizer (Kingma & Ba, 2014) by $lr_{base} \times (1 - e_{epochs}/E_{epochs})$.

On various image datasets named FashionMNIST (Xiao et al., 2017), CIFAR10 (Krizhevsky et al., 2009), and ImageNet (Russakovsky et al., 2015), we experimented with ResNet-based binarized CNNs. the FashionMNIST dataset has $28 \times 28$ color images with 10 classes, having 60K training and 10K test images. During training, no data augmentation was applied to the FashionMNIST dataset. On the other hand, the CIFAR10 dataset contains $32 \times 32$ color images with 10 classes, having 50K training and 10K test images. During training with data augmentation, $32 \times 32$ images were randomly cropped from $40 \times 40$ padded images and randomly flipped. The ImageNet dataset contains 1.3M training and 50K validation images with 1,000 classes. During training on the ImageNet dataset, $224 \times 224$ augmented images based on (Liu et al., 2020) were used. In inference, $224 \times 224$ center-cropped images from the validation dataset were adopted without the data augmentation. All experiments were conducted on an AMD Ryzen Threadripper PRO 5955WX 16-Cores CPU and 2 NVIDIA RTX 4090 GPUs and 263-GB RAM.

## A.3 Experiments on FashionMNIST and CIFAR10 datasets

During 400 epochs, the proposed OneBNet was trained with 256 mini-batch sizes. Label smoothing with $\epsilon = 0.2$ was adopted with zero weight decaying. Initial learning rate $lr_{base}$ was set as $1e-3$. We used ResNet20 and ResNet18 topologies as baseline models for the FashionMNIST and CIFAR10 datasets. Considering the image sizes and number of classes of each dataset, FashionMNIST and CIFAR10 adopted only ResNet20 and ResNet18 topologies, respectively.

Table 6: Summary of Top-1 accuracies on small-size datasets. Term *1-D Conv* means the number of output channels of 1-D binarized convolutional layers. Term *1-D DS* denotes the number of output channels of 1-D binarized convolutional layers in downsampling. If there is no mention with the terms, only 2-D binarized convolutional layers are adopted. The terms are used to indicate which layers adopted 1-D binarized convolutional layers.

| Dataset | Baseline | 1-D Conv | 1-D DS | FP32 (%) | OneBNet (%) |
|---|---|---|---|---|---|
| FashionMNIST | ResNet20 | -
64 | 32,64
64 | 93.6 | 93.4
93.2 |
| CIFAR10 | ResNet18 | -
256, 512 | 128,256,512
256, 512 | 95.5 | 93.6(TypeI)[1]
93.3(TypeII)[2] |

[1],[2] : terms TypeI and TypeII denote the model structures according to the usage of 1-D binarized convolutions.

Table 6 summarizes Top-1 accuracies and comparisons with FP32 counterparts. Terms *FP32 (%)* and *OneBNet (%)* denote Top-1 accuracies of FP32 counterparts and OneBNets. Top-1 accuracies from ResNet20 on the FashionMNIST dataset were 93.4% and 93.2%, where the accuracy drops were negligible, compared with FP32 ResNet20 model. On the CIFAR-10 dataset, TypeI and TypeII had 93.6% and 93.3% Top-1 accuracies, respectively.

Table 7: Summary of Top-1 accuracies by varying multiplier on CIFAR10 dataset. The multiplier $m$ is applied to the numbers of input and output channels.

| Dataset | Baseline | multiplier($m$) | Structure | OneBNet (%) |
|---|---|---|---|---|
| CIFAR10 | ResNet18 | 2 | TypeI
TypeII | 94.3
93.6 |
| | | 1.4 | TypeI
TypeII | 93.9
93.4 |
| | | 1 | TypeI
TypeII | 93.6
93.3 |
| | | 0.7 | TypeI
TypeII | 92.3
92.2 |
| | | 0.5 | TypeI
TypeII | 91.5
91.0 |

For verifying the performance varying on model size, we adopted the multiplier $m$ for the number of channels. Both the number of input and output channels can be scaled by multiplying with $m$. On the CIFAR10 dataset, the proposed OneBnet from baselined ResNet18 was evaluated by varying $m$. Whereas the computations

in convolutions can be scaled by $m^2$, those of element-wise operations are scaled by $m$. Table 7 shows the summary of Top-1 accuracies by varying multiplier on CIFAR10 dataset. In TypeI and TypeII, as $m$ increased, Top-1 accuracy slightly increased. Notably, when $m$ was 2, TypeI reached 94.3 Top-1 accuracy, which showed 1.2% degradation over its FP32 counterpart. When $m \geq 1$, the differences of Top-1 accuracy was not significant. However, $m < 1$, Top-1 accuracies were dramatically degraded in Table 7.

Table 8: Comparison of inference latencies using 4 threads on ImageNet dataset

| Model | Latency(ms) | Model | Latency(ms) |
|---|---|---|---|
| FP32 ResNet18 | 257.7 | XNOR-Net | 33.8 |
| Bi-RealNet | 31.3 | ReActNet18 | 40.3 |
| ReActNetA | 76.9 | **TypeI** | **52.4** |
| **TypeII** | **46.7** | **TypeIII** | **43.0** |

A.4 DETAIL DESCRIPTION OF EVALUATING INFERENCE SPEED

The target model was prepared using TensforFlow Keras framework. XNOR-Net (Rastegari et al., 2016), Real-to-Bin (Martinez et al., 2019), and Bi-RealNet (Liu et al., 2018) Keras models were from Larq Zoo. The models can be converted into a TFLite (TensorFlow Lite) filebuffer file by using Larq Compute Engine (LCE) (Bannink et al., 2021). When checking the inference speed of a model, we adopted Larq Compute Engine (Bannink et al., 2021), which provided a benchmark evaluation program based on TensorFlow Lite (ten, 2023) and the custom layers of binarized convolutional layers. It was known that LCE provided a collection of hand-optimized TFLite custom operators. Along with full supports of existing TFLite operators, a binarized convolutional layer can be converted into its custom binarized convolutional layer. In our evaluations, the program showed the averaged latencies of 150 runs with randomly generated inputs on RaspBerry Pi 4 (ras, 2023). where XNNPACK (xnn, 2023) was enabled.

We downloaded Manjaro 64-bit GNOME Desktop for Raspberry Pi 4. Then, a prebuilt binary to benchmark models was downloaded. When running the binary, several options can be chosen. We performed 150 runs, where 50 runs were repeated three times for suppressing the variation of achieving inference latencies. Whereas Table 3 was based on a single thread, Table 8 summarizes data based on **4 threads**. In this evaluation on 4 threads, it was noted that the latency gaps between baseline ReActNet18 and proposed models were reduced to 2.7ms by adopting more lightweight models.

We have added inference latency of ReActNetA (Liu et al., 2020) in Table 8. When using 4 threads on RaspberryPi 4, the inference latency of ReActNetA was 86.9ms on the ImageNet dataset. With a single thread, the inference latency was 123.3 ms. It is noted that ReActNetA was developed based on MobileNet (Howard et al., 2017). Although the estimated OPs from the binarized operations were small in ReActNetA, it showed longer latency.

A.5 OPERATIONS IN A BASIC BLOCK

---

**Algorithm 1** Operations in basic block

---

**Input:** $H \times W \times C_{in}$ activations

1: $residual1 = activations$
2: $learnable\ bias - sign$
3: $BinConv\ 3 \times 1 - BatchNorm$
4: **if** stride = 2 **then**
5: $\quad size\ of\ feature = (\frac{H}{2} \times W \times C_{out})$
6: **end if**
7: $shortcut(residual1)$
8: $residual2 = learnable\ bias - PReLU - learnable\ bias$
9: $learnable\ bias - sign$
10: $BinConv\ 1 \times 3 - BatchNorm$
11: **if** stride = 2 **then**
12: $\quad size\ of\ feature = (\frac{H}{2} \times \frac{W}{2} \times C_{out})$
13: **end if**
14: $shortcut(residual2)$
15: $learnable\ bias - PReLU - learnable\ bias$
16: **return** $output\ feature\ map$

---

Algorithm 1 describes the operations of a basic block in order. In the above algorithm, *learnable bias - sign* and *learnable bias - PReLU - learnable bias* denote RSign operation and RPReLU activation function (Liu et al., 2020). To adjust the activation distribution, RSign and RPReLU are invoked twice in the proposed basic block. Terms $shortcut(residual1)$ and $shortcut(residual2)$ refer to the summations of the outputs of 1-D binarized convolutional layers $BinConv3 \times 1$ and $BinConv1 \times 3$ and shortcuts $residual1$ and $residual2$, respectively. In other statements, the output of a previous layer is used in its next layer.

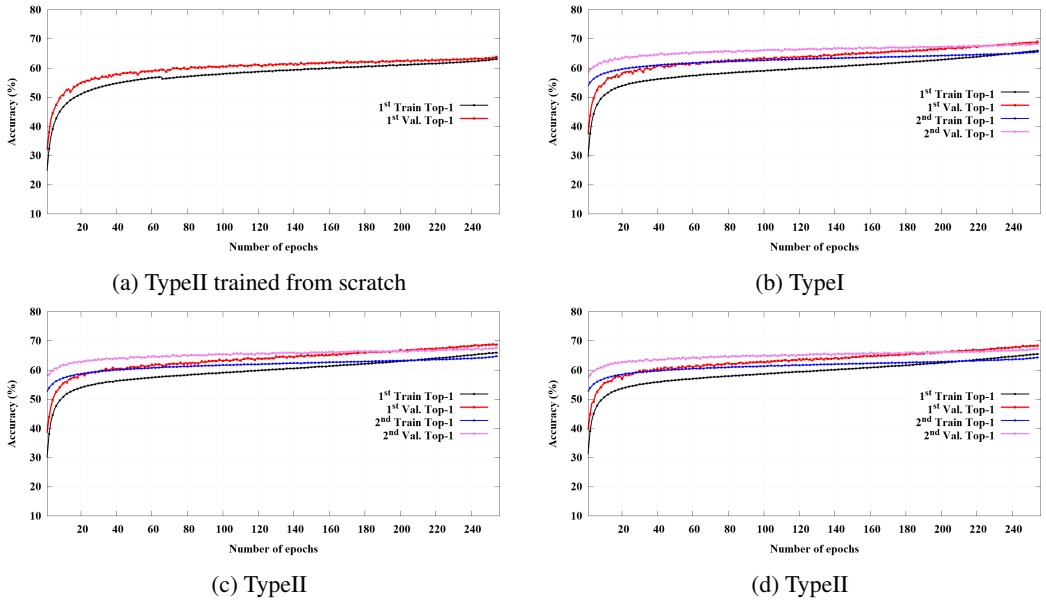

Figure 6: Training graphs of TypeI, TypeII, and TypeIII models. A term *scratch* denotes the training without using the two-stage training process and teacher-student training.

## A.6    TRAINING GRAPH ON IMAGENET DATASET

Figure 6 illustrates the training characteristics of TypeI, TypeII, and TypeIII during training epochs. When following the two-stage training setup, accuracies rapidly increased in the early stages with the Adam optimizer (Kingma & Ba, 2014). Using the data augmentation in Liu et al. (2020), validation accuracies were higher than training accuracies in each stage. Compared with the training from scratch, the gap between training and validation accuracies using the teacher-student training was greater in TypeII. Figure 6 shows the training can be well performed without overfitting on the ImageNet dataset.

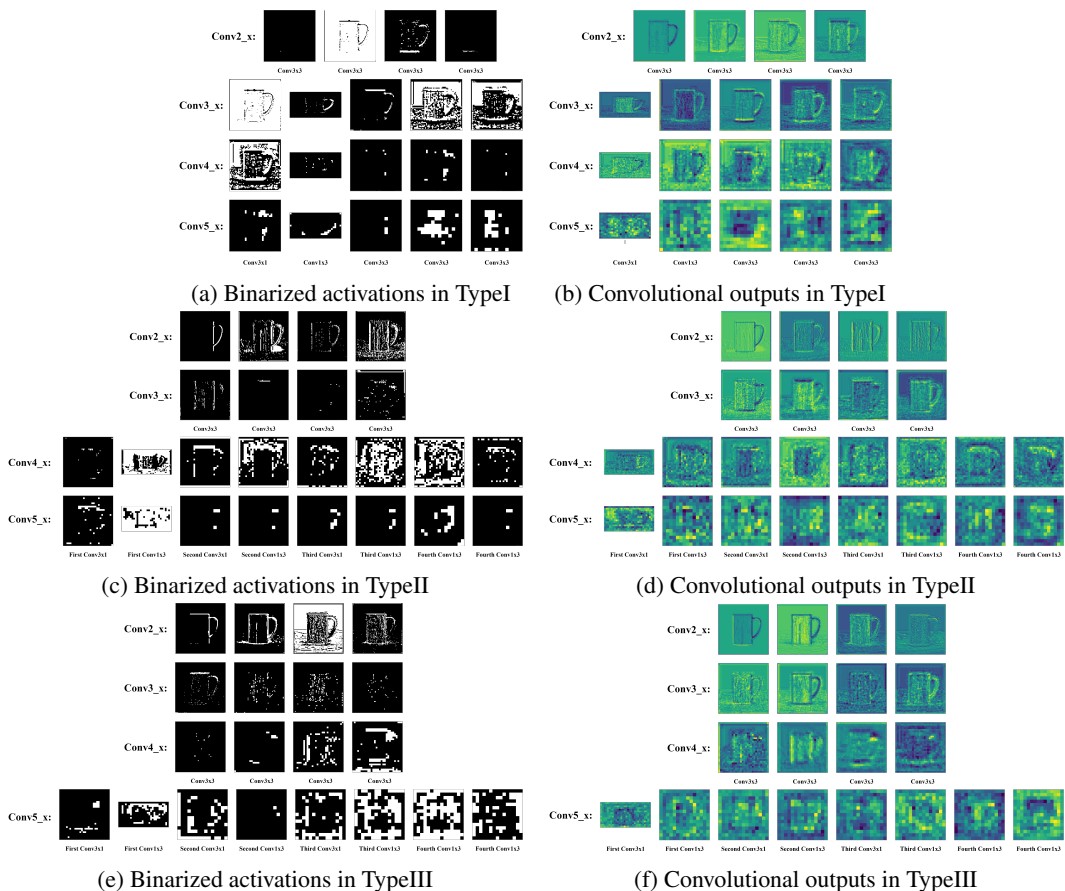

(a) Binarized activations in TypeI

(b) Convolutional outputs in TypeI

(c) Binarized activations in TypeII

(d) Convolutional outputs in TypeII

(e) Binarized activations in TypeIII

(f) Convolutional outputs in TypeIII

Figure 7: Illustration of activations in TypeI, TypeII, and TypeIII models.

## A.7 IMAGES OF ACTIVATIONS

Figure 7 illustrates several binarized activations and convolutional outputs in TypeI, TypeII, and TypeIII, which can be helpful to understand the image processing in the proposed OneBNet. Figure 7 shows that the image filtering using the proposed 1-D binarized convolutions was performed well. When it did not perform downsampling, TypeI adopted 2-D binarized convolutional layers. The rectangular images of Figure 7 represent the cases after the row-wise downsampling is performed. For example, **Conv1x3** of Figure 7 (a) illustrates the binarized activations after the row-wise $3 \times 1$ binarized convolutions were performed.

