# OpenReview forum: "OneBNet: Binarized Neural Networks using Decomposed 1-D Binarized Convolutions on Edge Device"
_ICLR.cc/2024/Conference — Submitted to ICLR 2024_

### Official Review · Reviewer_HiaQ · 2023-10-20

**Soundness:** 2 fair
**Presentation:** 2 fair
**Contribution:** 2 fair
**Rating:** 3
**Confidence:** 4

**Summary:**

This paper studied binarized convolutional neural network and proposed to replace nxn 2D BNN by two 1D BNN (nx1 row-wise BNN and 1xn column-wise BNN). Further, the paper designed two basic BNN blocks, 1-D binarized convolutional layer in Figure 1 and downsampling 1-D binarized convolutional layer in Figure 2. Adjustment of activation distribution are analyzed in Section 3.3. By combining above two basic blocks and adjustment of activation distribution, the paper built the architecture based on ResNet18 for image classification tasks.The experimental results validate the effectiveness of the proposed binarized convolutional neural network.

**Strengths:**

1. The paper provided detailed  analysis of proposed binarized convolutional network network from both theoretical and experimental perspectives.
2. The designed architectures achieved SOTA performances on CIFAR10 and imagenet.
3. The author's writing is very good, and the entire paper is relatively easy to understand.
4. simple algorithm, easy to follow.

**Weaknesses:**

1. from engineering viewpoints, the proposed basic blocks seems obviously due to 1）the work of binarizing nx1 and 1xn convolutional neural network existed, and 2) adjustment of activation distribution also existed. The combination of them is not novel enough for top conferences. Thus, the contribution of this method is not very important.
2. The reference format in this paper can be improved, e.g.
 - "Reactnet: Towards precise binary neural network with generalized activation functions" is an ECCV paper,
 - "Recu: Reviving the dead weights in binary neural networks" is an ICCV paper.
3. In provided tables, some columns are left aligned, while others are center aligned. It is best to use a consistent format

**Questions:**

see weaknesses above

---

> ### Author Response · Authors · 2023-11-16
> **Rebuttal for Review by Reviewer HiaQ**
>
> * **Question:**
> The works of binarizing $1 \times N$ and $N \times 1$ convolutional neural network and adjustment of activation distribution existed. The combination of them is not novel enough.
> * **Answer:**
> Thanks for your sharp comments. In this paper, the effects of adjusting the activation distribution are doubled on the decomposed 1-D binarized convolutions. Although 1-D binarized convolution was considered in an existing work (BNSC-Net), its performance was only under 60% Top-1 accuracy. When the ideas of 1-D convolution and the adjustments of activation distributions are used in the proposed OneBNet, we did not naively combine the ideas and performed the model developments and structural analysis as follows:
> **Firstly**, a new downsampling layer using 1-D binarized convolutions was proposed, so that the proposed model can have dramatical changes based on backboned ResNet, compared with the model naively using 1-D binarized convolution.
> **Secondly**, the effects of the 1-D binarized convolutions having the adjustments and shortcuts can be doubled. The detail structure for the doubled effects was proposed. To show which layers can be critical, model structures were analyzed in terms of Top-1 accuracy and latency. We showed that the proposed 1-D binarized convolutions in downsampling (denoted as 1-D DS in Table 1) can be effective in enhancing Top-1 accuracy. Besides, when the number of channels was great (256 or 512) and feature map size was small, the proposed model can be effective, which was explained in the previous draft as:
> ***"Table 1 shows the effectiveness of 1-D binarized convolutions in downsampling, where deep convolutional layers with many channels and small activation map can have significant benefits for enhancing performance."***
> ---
> * **Question:**
> The reference format in this paper can be improved.
> * **Answer:**
> In agreement with your concerns, the references of officially published papers have been changed in **Reference** section of the revised version as:
> ***"Zechun Liu, Zhiqiang Shen, Marios Savvides, and Kwang-Ting Cheng. Reactnet: Towards precise binary neural network with generalized activation functions. In European Conference on Computer Vision (ECCV), pp. 143-159, 2020."
> "Zihan Xu, Mingbao Lin, Jianzhuang Liu, Jie Chen, Ling Shao, Yue Gao, Yonghong Tian, and Rongrong Ji. Recu: Reviving the dead weights in binary neural networks. In Proceedings of the IEEE/CVF international conference on computer vision, pp. 5198-5208, 2021."***
> Besides, the references of several works have been revised if they were officially published.
> ---
> * **Question:**
> In provided tables, some columns are left aligned, while others are center aligned. It is best to use a consistent format.
> * **Answer:**
> According to your concerns, left-alignments have been applied to all Tables in the revised version.

---

> > ### Comment · Reviewer_HiaQ · 2023-11-21
> >
> > Thank you very much for your feedback and clarification. I agree that this paper has certain technical contributions and practical significance.
> >
> > I insist that the technology used in the paper mainly comes from existing literature, and the contribution of the technology combination lacks persuasiveness. For top-level conferences, current paper manuscripts have limited novelty and lack new insights. In addition, considering that other reviewers have already pointed out some modifications and clarifications, I believe that this article is not comprehensive enough.
> >
> > Therefore, I insist on my rating.

---

> > > ### Author Response · Authors · 2023-11-21
> > > **Answer for the second comments from Reviewer HiaQ**
> > >
> > > Thanks for your valuable comments. In agreement with your opinions, any theoretical new things and analysis will be additionally needed in the research of BCNNs. Your comments will be helpful to represent the main idea of this paper. Thanks again.

---

### Official Review · Reviewer_6xWT · 2023-10-30

**Soundness:** 3 good
**Presentation:** 3 good
**Contribution:** 3 good
**Rating:** 5
**Confidence:** 4

**Summary:**

This paper proposes a new binary neural network called OneBNet, which mainly replaces NxN 2D binarized convolution by Nx1 row-wise and 1xN column-wise 1D binarized convolutions. The proposed model shows strong performance on ImageNet, i.e., ResNet18-based model obtains 63.9% Top-1 accuracy by training from scratch and 68.4% Top-1 accuracy by applying teacher-student training.

**Strengths:**

+ The proposed method is easy to follow.
+ Latency on Raspberry Pi is reported, which is beneficial for the BNN community.
+ The proposed model shows strong performance on ImageNet, i.e., ResNet18-based model obtains 63.9% Top-1 accuracy by training from scratch and 68.4% Top-1 accuracy by applying teacher-student training.

**Weaknesses:**

-	It is not new to replaces NxN 2D binarized convolution by Nx1 and 1xN 1D binarized convolutions. For example, SqueezeNext (CVPR’18 workshop) decompose the KxK convolutions into two separable convolutions of size 1xK and Kx1. Although this paper focuses on binary neural network, the novelty of using such strategy for BNNs is still quite unclear.
-	Could the proposed method also suitable for object detection tasks using binary neural network?
-	Several recent methods reported in Table 2 are not included in Table 3.

**Questions:**

See the weakness part.

---

> ### Author Response · Authors · 2023-11-16
> **Rebuttal for Review by Reviewer 6xWT**
>
> * **Question:**
> It is not new to replaces $N \times N$ 2-D binarized convolution by $N \times 1$ and $1 \times N$ 1-D binarized convolutions.
> * **Answer:**
> Thanks for your novel comments. In agreements with your concerns, 1-D convolutions were proposed in SqueeNext and Inceptionv3. I agree that the naive application of 1-D convolutions to BCNNs cannot be a main contribution, as shown in **Introduction** section of the previous version.
> However, we provide several unique things in this paper as: the downsampling layer using 1-D binarized convolutions was proposed, so that the proposed model can have significant internal differences based on backboned ResNet, compared with the model naively using 1-D binarized convolution. Besides, the effects of adjusting the activation distribution and non-linear activation function can be doubled by applying specific layers for BCNNs to the decomposed 1-D convolutions in the proposed OneBNet. In order to avoid the confusion of the main contribution, the explanation has been rewritten in **Abstract** as:
> ***"It was known that the decomposed 1-D convolutions can replace the spatial 2-D convolutions in several convolutional neural networks (CNNs) for computer vision. However, the proper usage of 1-D convolutions was not shown in the field of binarized CNNs. This paper shows This paper proposes a new structure called OneBNet to maximize the effects of 1-D binarized convolutions, thus producing  excellent performance on CPU-based edge devices. To double the effects of adjusting the activation distribution and non-linear activation function, specific layers for BCNNs are doubled by applying them to  both $n \times 1$ row-wise and $1 \times n$ column-wise 1-D binarized convolutions. The proposed 1-D downsampling can perform information compression gradually through two 1-D convolutions, which can contribute tremendously to the performance improvement in binarized convolutional neural networks (BCNNs) in our analysis."***
> ---
> * **Question:**
> Could the proposed method also suitable for object detection tasks using binary neural network?
> * **Answer:**
> Thanks for your good recommendation for model applications. At this time, we did not verify the proposed OneBNet to object detection in the experiments. However, because similar works using BCNNs can produce acceptable performance in object detection, we expect that the proposed model could be applicable. For example, BiDet: An Efficient Binarized Object Detector (CVPR 2020) and Recurrent Bilinear Optimization for Binary Neural Networks (RBONN in ECCV 2022) applied BCNNs such as XNOR-Net, Bi-RealNet, and ReActNet in the object detection. Notably, when RBONN was applied to the image classification, its Top-1 performance (66.7\%) on ImageNet dataset cannot reach our results.
> Although the experimental results for the object dection were not listed,
> the successful application with the counterparts and the outstanding image classification of the proposed model make sure that the proposed OneBNet could be useful in object detection.
> ---
> * **Question:**
> Several recent methods reported in Table 2 are not included in Table 3.
> * **Answer:**
> Table 3 listed the comparison with several BCNNs if a model has difference in layer structure and **the specific layer supported by Larq compute engine**.
> When the specific layer was not supported by Larq, we cannot evaluate the latency using Larq. For example, XNOR-NET++ has row-wise and column-wise scaling parameters in binarized convolutions, which cannot be supported in Larq and Tensorflow Lite. In RB-Net, the reshaped output could not be supported by Larq. Therefore, we have listed models that can be developed by Larq in Table 3.

---

> > ### Comment · Reviewer_6xWT · 2023-11-22
> > **Thanks for the response**
> >
> > Thanks for the detailed response. Overall my concerns about limited novelty and more experiments on downstream tasks have not been fully addressed, thus I tend to keep my rating.

---

### Official Review · Reviewer_w6Pw · 2023-10-30

**Soundness:** 1 poor
**Presentation:** 3 good
**Contribution:** 1 poor
**Rating:** 3
**Confidence:** 4

**Summary:**

This paper proposes to decompose a 2D convolution into two 1D convolutions on a Binarized neural network model to improve inference speed and model accuracy on edge devices. On the basis of the previous Binarized ResNet, the 3x3 convolutional kernel was changed to two sets of 1x3 and 3x1 convolutional kernels, achieving higher accuracy.

**Strengths:**

This paper is easy to read and understand. The method in this paper is relatively simple, clear, and easy to reproduce. The experimental data in the paper indicates that the improved model outperforms the previous Binarized neural network model in terms of speed and accuracy.

**Weaknesses:**

1. The contribution and innovation of this paper are insufficient. The decomposition of 2D convolutions into two 1D convolutions used in this article is not a new idea, but a widely studied method. Although its combination with Binarized neural networks may make it more effective, it is easy to consider or attempt.

2. The generalizability of this method has not been verified. The author's experiment only trained and tested the smaller ResNet model, and the dataset only included CIFAR10 and ImageNet.

3. The basic theory of the method in this paper is not sound enough. Binary quantization and 2D convolutional decomposition are methods that sacrifice accuracy for less computational complexity. Why can a combination of the two achieve better accuracy? Substantive improvements may come from element wise calculations and learnable bias in more layers after decomposition, but this is not without cost, as FLOPs cannot accurately reflect the additional hardware overhead this brings. The explanation of Figure 3 also lacks quantitative data support.

4. This method lacks a determination method for parameter selection. The paper mentions that not all convolutional layers of blocks are suitable for such transformations. In Table 1, several selection combinations are attempted, and the best ones are selected for subsequent comparison. This will bring difficulties to practical applications. If the model structure is different and there are more blocks with different channel numbers, it will not be suitable for such selection.

5. The comparison of experiments lacks fairness and universality. Compared to other methods in Table 3, they are all different binary quantization methods and will not significantly change the model structure. This paper essentially changes the structure of the model orthogonal to previous work. Unless compared with other structural optimizations, such changes are unfair.

**Questions:**

In experimental environments, it said "Like other BCNN models, the first convolutional and last fully-connected layers adopted FP32
weights and activations." Is the latency data measured End2End? If not, are non binarized layers becoming performance bottlenecks?

---

> ### Author Response · Authors · 2023-11-16
> **Rebuttal for Review by Reviewer w6Pw**
>
> * **Question:**
> The contribution and innovation of this paper are insufficient.
> * **Answer:**
> In agreements with your concerns, 1-D convolutions were proposed in previous works such as SqueeNext and Inceptionv3. I agreed that the naive application of 1-D convolutions to BCNNs cannot be a main contribution, as shown in Introduction section.
> On the other hand, our works proposed the downsampling layer using 1-D binarized convolutions, so that the proposed model can have significant internal changes. Besides, the effects of adjusting the activation distribution and non-linear activation function can be doubled by applying specific layers for BCNNs to the decomposed 1-D convolutions. To focus on the main contribution, the explanation has been rewritten in **Abstract** as:
> ***"It was known that the decomposed 1-D convolutions can replace the spatial 2-D convolutions in several convolutional neural networks (CNNs) for computer vision. However, the proper usage of 1-D convolutions was not shown in the field of binarized CNNs. This paper proposes a new structure called *OneBNet* to maximize the effects of 1-D binarized convolutions, thus producing excellent performance on CPU-based edge devices. To double the effects of adjusting the activation distribution and non-linear activation function, specific layers for BCNNs are doubled by applying them to both $n \times 1$ row-wise and $1 \times n$ column-wise 1-D binarized convolutions. The proposed 1-D downsampling can perform information compression gradually through two 1-D convolutions, which can contribute tremendously to the performance improvement in binarized convolutional neural networks (BCNNs) in our analysis.''***
> ---
> * **Question:**
> Generalizability: Only smaller ResNet model, and the dataset only included CIFAR10 and ImageNet.
> * **Answer:**
> Our evaluations were based on baselined ResNet18 and ResNet20. Target datasets included FashionMNIST, CIFAR10, and ImageNet. In agreement with your concerns, we adopted the multiplier $m$ for the number of channels in Appendix A.3 of the revised version, . Both the number of input and output channels can be scaled by multiplying with $m$. New Table 7 shows the summary of Top-1 accuracies by varying multiplier on CIFAR10 dataset.
> ---
> * **Question:**
> Why can a combination of Binary quantization and 2D convolutional decomposition achieve better accuracy?
> * **Answer:**
> Because the newly added adjustments and functions are doubled, we concluded that the doubled effects can mitigate the degradation. I agree that good performance of the proposed method required additional computational costs. Therefore, we want to show the trade-off between the additional costs and enhanced performance to provide better model structure, in Figure 5 and Table 1.
> We agree with you that OPs in Figure 5 cannot be accurate to estimate the latencies. Additionally, Table 1 compares the latencies on Raspberry Pi when1-D binarized convolutions were adopted, so that the guideline for developing structures were concluded in terms of accuracies and latencies.
> ---
> * **Question:**
> This method lacks a determination method for parameter selection.
> * **Answer:**
> Although it is hard to extract any quantitative metric, we think that the below guideline can introduce the priority to apply the proposed structure. In the second paragraph of **Experimental Results and Analysis** section, we concluded the meaning of data as:
> ***"Table 1 shows that the effectiveness of 1-D binarized convolutions in downsampling, where deep convolutional layers with many channels and small activation map can have significant benefits for enhancing performance."***
> The above conclusion gave the guideline to determine which layer adopted 1-D binarized convolutions considering latencies and Top-1 accuracies.
> ---
> * **Question:**
> In Table 3, unless compared with other structural optimizations, such changes are unfair.
> * **Answer:**
> The proposed OneBNet adopted 1-D binarized convolutions, which bring structural changes. However, many characteristics based on the baselined ResNet were equally applied; the first FP32 convolutional and last fully-connected layers were the same in Table 3. the numbers of downsamplings and output channels in their basic blocks can be the same. In several existing works such as Inceptionv3 and BNSC-Net, the computations and accuracies of the structures having decomposed 1-D convolutions are compared with those of models using 2-D convolutions. Besides, we think that the differences of the latencies in Table 3 are due to the structural changes using the specific layers. For example, Real-to-Bin had self-attention layer. Whereas XNOR-Net adopted doubled skipped shortcuts, other BCNNs had single skipped shortcuts. In **Related Works** section, the model structures were briefly summarized.
> ---
> * **Question:**
> Is the latency data measured End2End?
> * **Answer:**
> The latencies included the delay in the first and fully connected layers.

---

> > ### Comment · Reviewer_w6Pw · 2023-11-17
> >
> > Thank you for your reply and explanation to my question. Inserting more adjustment layers does bring benefits. However, I believe that the overall novelty of this paper **is still limited**, as almost all components come from existing research.
> >
> > I think it is possible to revise my rating to 5 if more test data is provided on other baseline models or different types of tasks. This is because I agree that the author's current work has good practical significance, but it is still not sufficient as a good paper with in-depth insights or new contributions.
> >
> > In addition, 1D convolutional decomposition is not the core contribution point, so I think an interesting perspective is the effectiveness and universality of inserting adjustment layers after this decomposition. As you mentioned, *"Table 1 shows that the effectiveness of 1-D binarized conversions in downsampling, where deep convolutional layers with many channels and small activation map can have significant benefits for enhancing performance"*
> > For the Binary quantization model, the rule of inserting adjustment layers after decomposition calculation is applicable to ResNet, but is it applicable to the Inception, MobileNet structure or even Transformer? If so, this will significantly increase the significance of the work.

---

> > > ### Author Response · Authors · 2023-11-18
> > > **Answer for the second comments from Reviewer w6Pw**
> > >
> > > Thanks for your valuable comments.
> > > Although the components of the proposed model are used in other models,
> > > we think that the appropriate deployments in BCNNs for outstanding performance is essential.
> > > Existing developments in many BCNNs adopted the same way as ours.
> > > For example, Real-to-Bin (ICLR 2020) adopted self-attention block to BCNNs.
> > > ReActNet (ECCV 2020) used teacher-student training and transfer learning to BCNNs.
> > > The contribution points in those papers had been used in the field of FP32 model.
> > > But they showed that the block and training method are very effective.
> > > Besides, the commercial supports for the approach and easy application are now avaliable.
> > > The function of learnable parameters can be an affine layer.
> > > The goal of our research is how to deploy the components,
> > > having dramatic advance in performance and small additional overhead
> > > on practical environments.
> > > Most of all, this paper can provide an advanced point for achieving the goal,
> > > reducing performance gap from FP32 baseline.
> > > We think that our approach can advance the performance of BCNNs
> > > with easy applicable configurations.
> > >
> > >
> > > ---
> > > The answer for the second question is as follows:
> > > When applying 1-D binarized convolutions and additional element-wise layers,
> > > $3 \times 3$ 2-D convolutional layers can be targeted.
> > > From our internal researches, the residual network with shortcut is necessary due to large quantization error from binarization.
> > > The original Inception, MobileNet, and Transformer should be modified to have shortcuts.
> > > I think that the binarization of Inception and MobileNet can be done after modifying them.
> > > However, element-wise operations can increase latency,
> > > so that additional analysis will be needed to find the optimal structure.
> > > Notably, ReActNetA (ECCV 2020) modified MobileNetv1,
> > > where 2-D binarized conventional convolutions replaced $3 \times 3$ deptwise separable convolutions.
> > > We think that because the modified version has $3 \times 3$ convolutions,
> > > our structure can be applicable.
> > > Instead of conventional Vision Transformer,
> > > Mobile Vision Transformer(ViT) such as MobileViT adopted 2-D convolutional layers.
> > > The binarization can be applicable.
> > > However, the performance of mobile ViT should be advanced,
> > > compared to FP32 Transformer.
> > > Besides, because of long latency of mobile ViT,
> > > the straightforward comparison with BCNNs seems not to be reasonable.

---

### Official Review · Reviewer_38Cc · 2023-10-31

**Soundness:** 2 fair
**Presentation:** 1 poor
**Contribution:** 2 fair
**Rating:** 3
**Confidence:** 3

**Summary:**

The work proposes decomposing 2-D binarized convolution into two 1-D convolutions in different directions to reduce model complexity. Then, the paper investigates the settings where such replacement can be beneficial, followed by experimental verification of the architecture performance.

**Strengths:**

This work identified the proper settings where replacing 2D convolutions with 1D convolutions can be beneficial. The model performance shown in the paper is promising.

**Weaknesses:**

Overall the paper should improve on the presentation befoer it is ready for publication. At current state, I am unsure about many technical details. Please also see below.

**Questions:**

P3: section 3.1 The section is very difficult to follow, please rewrite this.

P4: "However, it shrinks the receptive fields", I am not sure whether this is true, please verify.

Fig. 2. It is not clear how the results from 1D convolution are summed together with ouput of the 1x1 FP32 convolution, given they are of different shape.

P5: not sure about " It removes the negative part with β"

P5: Not sure why there should be both  ζ and α

Figure 5: please change Fig. (a) to (a)

Table 1 is very confusing, please clear it up.

Distillation is quite a standard approach for improving model performance. Therefore, I suggest the authors move the results from the distillation experiment to the appendix.

I suggest adding a model performance comparison with a similar OP level.

ReActNet presented results of much higher performance (probably higher complexity); I suggest including an experiment that compares those results.

---

> ### Author Response · Authors · 2023-11-16
> **Rebuttal for Review by Reviewer 38Cc**
>
> Thanks for your review. We have addressed your concerns and upload new PDF. Please check them.
> * **Question:**
> Section 3.1 is very difficult.
> * **Answer:**
> According to your recommendation, **Motivations** section has been modified.
> Notably, the second paragraph was rewritten as:
>  **"*To double the effects of the adjustment of activation distribution, we apply the adjustment of activation distribution to the decomposed 1-D binarized convolutions. Whereas the conventional 2-D convolutions use 3 × 3 kernels, the decomposed convolutions use 3 × 1 and 1 × 3 kernels. The adjustment of activation distribution is applied to each 1-D convolution. Although the decomposition had the same receptive field with 2-D convolution (Szegedy et al., 2016), it requires additional element-wise operations for the adjustments and activation functions. This paper explains the structure of 1-D convolutional layer. Then, it analyzes the model structures in terms of latency and accuracy and shows the idea to deploy 1-D binarized convolutions on the above explained pros and cons*."**
> ---
>
> * **Question:**
> I am not sure "However, it shrinks the receptive fields"  is true.
> * **Answer:**
> According to Inceptionv3, when $1 \times n$ and $n \times 1$ convolutions are used, the receptive fields are the same. The expression has been modified as:
> ***"Although the decomposition had the same receptive field with 2-D convolution (Szegedy et al., 2016), it shrinks the receptive fields and requires additional element-wise operations for the adjustment."***
> ---
> * **Question:**
> Fig. 2 needs to be clarified.
> * **Answer:**
> The numbers of channels and size of the feature map after 1x1 FP32 and binarized convolutions have been denoted in Fig.2.
> ---
> * **Question:**
> The expression "It removes the negative part with $\beta$"
> * **Answer:**
> The expression has been rewritten as:
> ***"When $x_i - \gamma_i$ is negative, learnable parameter $\beta_i$ scales it."***
> ---
> * **Question:**
> Not sure why there should be both $\zeta$ and $\alpha$.
> * **Answer:**
> The learnable parameter $\alpha$ can produce the difference between shortcut and input of binarized convolutions. If the learnable parameter does not exist, the input activations for the shortcut and binarized convolutions are the same. The learnable parameter can adjust the input distribution for the binarized convolution. Although $\zeta$ can change the distribution for the shortcut and input of binarized convolutions, it is known that the additional adjustments with $\alpha$ are necessary in the ablation study of ReActNet (Liu et al., 2020).
> ---
> * **Question:**
> In Figure 5, change Fig. (a) to (a)
> * **Answer:**
> We have revised the expression in the caption of Figure 5.
> ---
> * **Question:**
> Table 1 is very confusing.
> * **Answer:**
> To avoid confusion in Table 1, the inference latencies of all structures have been listed in the revised version.
> ---
> * **Question:**
> I suggest adding a model performance comparison with a similar OP level.
> * **Answer:**
> To show the comparison with BCNN models having similar OPs, counterparts listed in Table 3 were compared in the previous manuscript because they have the same first FP32 convolutional and last fully connected layers on the same baseline model.
> Instead, I think that your recommendation means the comparison with mobile-friendly FP32 models such as MobileNet or ShuffleNet. Theoretically, TypeI had about **$170 \times 10^6$** OPs. On the other hand, MobileNetv1 and MobileNetv2 has about **500MFLOPs and 300MFLOPs** with 70.6 and 72.0 Top-1 accuracies. The inference latency on a single thread of MobileNetv2 was **117ms** using Larq. Considering the results, we can conclude that the proposed OneBNet can consume small computational resources, although accuracies were degraded.
> The performance comparison with other mobile-friendly models requires huge space because there have been too many researches. Besides, additional model optimization related to number of channels and feature size should be done. Thanks for recommendation.
> ---
>
> * **Question:**
> Experiment that compares those results with ReActNet.
> * **Answer:**
> Thanks for the good recommendation. ReActNet and its following works based on had better Top-1 accuracy, but the models were based on a modified MobileNet. Although the theoretical OP of ReActNetA was smaller than BCNNs based on ResNet18, our experiment shows that its latency was significantly longer than the proposed model. We have added the experimental results and discussion in **Appendix A.4** as:
> ***"We have added the inference latency of ReActNetA (Liu et al., 2020) in Table 7. When using 4 threads on RaspberryPi 4, the inference latency of ReActNetA was 86.9ms on the ImageNet dataset. With a single thread, the inference latency was 123.3 ms. It is noted that ReActNetA was developed based on MobileNet (Howard et al., 2017). Although the estimated OPs from the binarized operations were small in ReActNetA, it showed longer latency."***

---

### Comment · Area_Chair_Ns95 · 2023-11-23
**[ICLR 2024 Reviewers’ feedback] Please read authors’ responses and give your feedback**

Dear Reviewers,

Thanks again for your strong support and contribution as an ICLR 2024 reviewer.

Please check the response and other reviewers’ comments. You are encouraged to give authors your feedback after reading their responses. Thanks again for your help!

Best,

AC

---

### Meta-Review · Area_Chair_Ns95 · 2023-12-12

**Metareview:**

All reviewers gave negative scores. Although the authors responded to each reviewer, all reviewers kept the scores.

The paper provided a detailed analysis of the proposed binarized convolutional network. This paper has certain technical contributions and practical significance. The writing is good and easy to follow.

The overall novelty of this paper is still limited, as almost all components come from existing research. The paper is not yet ready for publication in its current state.

The authors are encouraged to further polish the paper by considering reviewers' comments.

**Justification For Why Not Higher Score:**

The rebuttal did not address the reviewers' concerns well. All reviewers keep negative ratings.

**Justification For Why Not Lower Score:**

N/A

---

### Decision · Program_Chairs · 2024-01-16

Reject